# Features of Liver Injury in COVID-19 Pathophysiological, Biological and Clinical Particularities

Cristina Maria Marginean [1] [ID], Eliza Cinteza [2,3] [ID], Corina Maria Vasile [3,4,*] [ID], Mihaela Popescu [5,*],
Viorel Biciusca [1] [ID], Anca Oana Docea [6], Radu Mitrut [7], Marian Sorin Popescu [8] and Paul Mitrut [1]

[1] Department of Internal Medicine, University of Medicine and Pharmacy of Craiova, 200349 Craiova, Romania
[2] Pediatrics Department, University of Medicine and Pharmacy "Carol Davila", 020021 Bucharest, Romania
[3] Department of Pediatric Cardiology, "Marie Curie" Emergency Children's Hospital, 041451 Bucharest, Romania
[4] Department of Pediatric and Adult Congenital Cardiology, Bordeaux University Hospital, 33600 Pessac, France
[5] Department of Endocrinology, University of Medicine and Pharmacy of Craiova, 200349 Craiova, Romania
[6] Department of Toxicology, University of Medicine and Pharmacy of Craiova, 200349 Craiova, Romania
[7] Department of Cardiology, University and Emergency Hospital, 050098 Bucharest, Romania
[8] Ph.D. School Department, University of Medicine and Pharmacy of Craiova, 200349 Craiova, Romania
* Correspondence: corina.vasile93@gmail.com (C.M.V.); mihaela.n.popescu99@gmail.com (M.P.)

**Abstract:** The outbreak of the coronavirus pandemic in March 2020 has caused unprecedented pressure on public health and healthcare. The spectrum of COVID-19 onset is large, from mild cases with minor symptoms to severe forms with multi-organ dysfunction and death. In COVID-19, multiple organ damage has been described, including lung damage, acute kidney injury, liver damage, stroke, cardiovascular and digestive tract disorders. The aspects of liver injury are different, sometimes presenting with only a slight increase in liver enzymes, but sometimes with severe liver injury, leading to acute liver failure requiring liver transplantation. In patients with chronic liver disease, especially liver cirrhosis, immune dysfunction can increase the risk of infection. Immune dysfunction has a multifactorial physiopathological mechanism, implying a complement system and macrophage activation, lymphocyte and neutrophil activity dysfunction, and intestinal dysbiosis. This review aims to evaluate the most relevant studies published in the last years related to the etiopathogenetic, biochemical, and histological aspects of liver injury in patients diagnosed with COVID-19. Liver damage is more evident in patients with underlying chronic liver disease, with a significantly higher risk of developing severe outcomes of COVID-19 and death. Systemic inflammation, coagulation disorders, endothelial damage, and immune dysfunction explain the pathogenic mechanisms involved in impaired liver function. Although various mechanisms of action of SARS-CoV-2 on the liver cell have been studied, the impact of the direct viral effect on hepatocytes is not yet established.

**Keywords:** COVID-19; liver injury; liver enzymes; SARS-CoV-2; hepatic lesions

## 1. Introduction

COVID-19 clinical presentation falls on a wide spectrum, from mild cases complaining of minor symptoms to severe illness with multiorgan dysfunctions and death. Multiple organ injuries have been described in COVID-19, such as pulmonary affliction, acute kidney damage, liver injury, stroke, cardiovascular and digestive tract disorders [1,2]. The literature recognizes the threat of hepatocyte infection and subsequent hepatic injury, the virus using angiotensin 2 converting receptor protein (ACE2) to infiltrate the cells. The aspects of liver injury are also large, sometimes manifested only with a mild increase in liver enzymes, but sometimes with severe liver injury, determining acute liver failure that requires liver transplantation [3].

This literature review aims to outline some of the most important and recent aspects of COVID-19 infection on liver function based on the comprehensive data reported since the pandemic outbreak.

## 2. Materials and Methods

The purpose of this review is to make a comprehensive and integrated approach to the essential aspects of liver involvement in COVID-19 disease, including pathogenic mechanisms, biochemical abnormalities, and correlations between preexisting liver diseases and effects of SARS-CoV-2, researching the most exhaustive publications in the current literature.

A series of electronic searches in PubMed was conducted using the following keywords: "COVID-19", "SARS-CoV-2", "chronic liver disease", "cirrhosis", "drug-induced liver injury", and "NAFLD". Articles obtained from these search queries were manually assessed for information quality and results significance. References cited in selected papers were also evaluated and included if deemed relevant.

## 3. Pathophysiology and Histology

In SARS-CoV-2 replication, tissue reservoirs are not completely elucidated due to difficulties in collecting biopsy samples and in isolating the samples in high-level laboratories.

SARS-CoV-2 binds to ACE2 to penetrate host cells [3]. Transmembrane protease serine 2 (TMPRSS2) and essential amino acid cleavage enzyme (FURIN) are also crucial in the production of the infection [4]. ACE2 cellular entry receptor is widely expressed in human tissues such as in the lungs (predominantly type II alveolar cells), gastrointestinal tract cells (esophageal epithelium cells, also enterocytes), liver (hepatocytes and cholangiocytes), cardiovascular system (myocardial cells), kidney (proximal tubular cells and urothelium), as well as in the pancreas [5].

Recent studies have found that ACE2 expression in cholangiocytes is significantly increased versus in hepatocytes (59.7% vs. 2.6%) [6], which suggests that SARS-CoV-2 determines a direct cytopathic effect by binding to cholangiocytes that express ACE2. Cholangiocytes have a definite role in hepatocyte regeneration mechanisms and immune response; thus, alteration of their function can cause hepatobiliary lesions, an aspect suggested by an increased titer of enzymatic cholestasis markers (gamma-glutamyl transferase) [7–9].

The histological changes in patients infected with SARS-CoV-2 and underlying liver disease have not yet been established. However, a significant increase in ACE2 expression in the liver of patients with chronic hepatitis C virus infection (HCV) compared to healthy individuals is documented in previous studies. The expression of ACE2 in human cirrhosis liver was described in 2005 by Paizis et al., with researchers detecting the receptor in most hepatocytes contained inside cirrhotic nodules and endothelial cells [10]. Pre-existing liver damage and inflammation appear to potentiate SARS-CoV-2 liver tropism by modulating ACE2 receptor expression [11,12].

SARS-CoV-2 PCR is positive in stool samples until one week after viral lung elimination [13–15]. Enterocyte infection was objected to by identifying RNA and viral proteins that persist in the intestinal mucosa for several months after recovery [16].

There is no clear evidence of direct specific liver tropism of SARS-CoV-2 [17], although liver histological lesions induced by SARS-CoV-2 cannot be denied.

Vascular abnormalities such as venous and sinusoidal micro thrombosis (100%), micro and macrovesicular steatosis (50%), mild inflammation of the portal spaces (66%), and portal fibrosis (60%) were histologically evidentiated in several studies, which suggests the presence of pre-existing liver disease, especially non-alcoholic hepatic steatosis (NASH), with or without associated metabolic risk factors, such as hypertension or impaired glucose tolerance.

It was postulated that the liver is a target for SARS-CoV-2 adhesion, because the similarity between ACE2 expression in alveolar cells and cholangiocytes [18,19] and hepatocyte degeneration, focal necrosis, and canalicular cholestasis were observed in a microscopic

study [20]. Glycogen accumulation in hepatocytes, sinusoidal dilatation, moderate lymphocytic infiltration, and necrosis were also described in the centrolobular areas [19,21].

Underlying liver disease, hepatotoxic drugs, and hyperinflammatory status due to SARS-CoV-2 infection can determine moderate to severe histological liver lesions favored by tissue hypoxia (Figure 1).

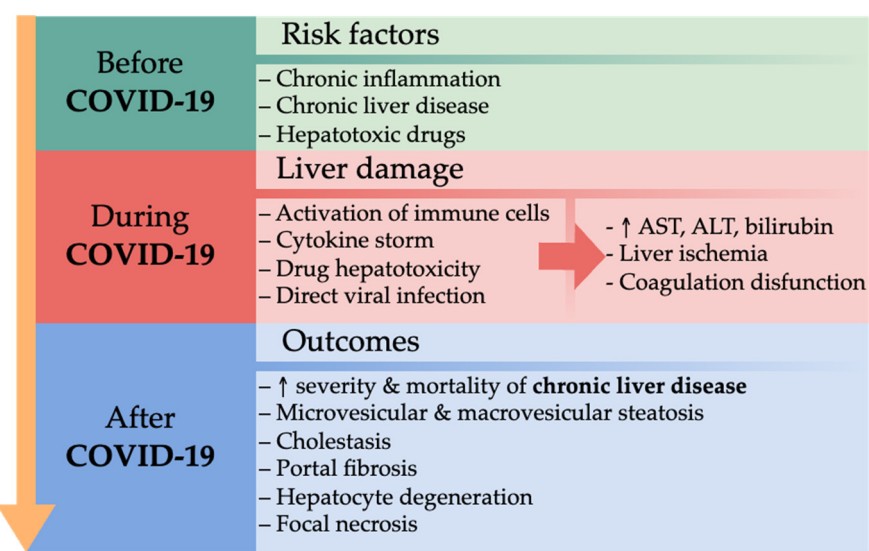

**Figure 1.** Development of COVID-19-induced liver lesions.

## 4. Biochemical Abnormalities in COVID-19

Hepatic biochemical abnormalities are commonly encountered in patients with COVID-19. Their incidence is between 15% and 65% of people infected with SARS-CoV-2, according to recently published studies [22–30].

Alterations in liver biochemistry in COVID-19 consist of generally slight increases (×1–2 more than the upper limit) in serum levels of alanine aminotransferase (ALT) and aspartate aminotransferase (AST) [23–25,31]. These appear both in previously healthy subjects and individuals with pre-existing hepatic injury [28]. Based on the results from the study conducted by Fu et al., patients with raised AST levels and total bilirubin (TBIL) had an unfavorable prognosis [25].

Another study by Hajifathalian et al. on a target group of 1059 patients diagnosed with COVID-19 showed that 62% had a minimum of one elevated liver enzyme [32]. Similar results were reported based on the study conducted by Cholankeril et al. on a smaller group of 115 patients [33].

High levels of AST and GGT were not associated with survival rates, according to Bernal-Monterde et al.'s retrospective study, on a lot of 540 subjects [34].

Various factors are responsible for biochemical anomalies, with possible contributing mechanisms being the immune-mediated inflammatory response, liver congestion, drug-induced liver damage, and direct liver cell infection [35]. In hospitalized patients, increased AST correlates positively with ALT but not with CRP (C-reactive protein), ferritin, or creatine kinase [7], suggesting that COVID-related liver enzyme elevation is a result of direct liver injury and that systemic inflammatory syndrome may be associated [36].

In most cases, AST is more increased than ALT, atypical for the classic model of hepatocellular lesions outside known circumstances, such as alcoholic liver disease or specific drug-induced liver lesions [7]. The responsible mechanisms that determine a predominant increase in AST in COVID-19 are incompletely studied and can be attributed to hepatic steatosis induced by SARS-CoV-2 [29], mitochondrial dysfunction associated with COVID-19 [30,35], as well as by hepatic perfusion impairment due to micro-thrombotic status [37,38]. Increased AST also correlates with systemic hypoxia, affecting patients with COVID-19 [39,40].

Increased liver enzymes are also correlated with the release of proinflammatory cytokines [41], with a significant increase in CRP, D-dimers, IL-6, and serum ferritin being described [8,31,42,43]. Increased levels of IL-6 are associated with liver damage [27,44]. The serum concentration of IL-6 correlates with the severity of the infection, increasing during the disease and decreasing post-recovery [45].

Hepatocytic ischemia, tissue congestion, and hepatic arteriovenous thrombosis [37,44] are the main factors that alter hepatic biochemical tests [46–49].

The predictive value of elevated liver enzymes in patients with SARS-CoV-2 infection is continuously studied. Some studies have shown correlations between increased serum liver enzymes and severe disease, sometimes requiring admission to intensive care and mechanical ventilation [30,50–54]. In contrast, other studies have reported no clear association between raised hepatic enzymes and mortality [52,55]. A study by Mohamed et al. highlighted statistically significant severe outcomes in patients with abnormal liver function and histopathological lesions [56].

The prognostic significance of abnormal hepatic biochemistry could be correlated with the host's immune response and the use of aggressive therapies in severe patients [24,31,57–59].

## 5. Drug-Induced Liver Damage

Drug-induced liver damage (DILI) is generally rare but represents a significant cause of acute liver failure with high mortality; many drugs can cause it, and differential diagnosis is difficult.

Hospitalized COVID-19 patients need complex treatment [60]. In this context, concerted studies on pharmacological agents' hepatotoxicity are essential in diagnosis because drug hepatotoxicity may vary depending on patients' race, age, or gender [61].

Many drugs can impair liver function; some may cause an asymptomatic increase in liver enzymes; in other cases, they determine acute liver failure.

Drug-induced liver injury most commonly increases liver enzymes due to experimental antiviral therapies (Figure 2) [49]. Liver damage caused by antivirals such as lopinavir–ritonavir [50,62] and remdesivir has been studied, with the hepatotoxicity being extensively studied. Antibiotics, antivirals, and anti-inflammatories, paracetamol, and tocilizumab [63,64] can cause liver damage [65].

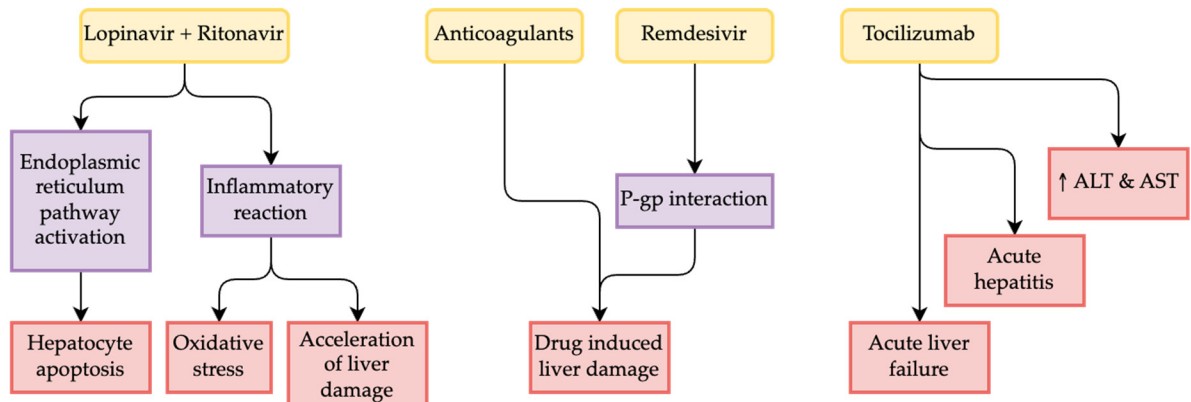

**Figure 2.** Drug hepatotoxicity due to COVID-19 treatment.

Combining antivirals with a risk of overdose (Ritonavir and Lopinavir) can induce hepatocyte apoptosis by activating the endoplasmic reticulum course by the caspase cascade system, causing oxidative stress through consecutive inflammatory reactions. A study conducted by Cai et al. on 417 patients from Shenzhen, China, reported that antivirals are correlated with $4\times$ increased risk of hepatic damage [51].

Low molecular-weight heparin (LMWH) liver damage is an uncommon and reversible adverse effect. The mechanism is unknown, with a possible idiosyncratic effect being described [66]. LMWH has been associated with an elevation of ALT and AST in 4–13% of

patients. However, values over 5× the upper limit of normal are uncommon and appear in individuals treated with high doses. ALT/AST usually increases within 3–7 days of anticoagulant treatment and are generally moderate or asymptomatic, improving quickly after discontinuing anticoagulant therapy. Liver enzyme values often decrease even when anticoagulant administration is continued in therapeutic doses [64].

Generalized inflammation determined by the activation of the cytokine cascade can determine multiorgan dysfunction and severe complications in patients with SARS-CoV-2 infection, with both pulmonary and cardiac or liver damage. IL-6 inhibitors, such as tocilizumab used in COVID-19 to reduce hyperactive inflammation, can determine severe liver damage, such as acute hepatitis or acute liver failure, requiring the need for liver transplant [67].

Liver injury incidence varies by drug but registers an increased parallel to the number of agents administered.

Colchicine, medication used in COVID-19 patients to reduce inflammatory status [68,69], is associated with liver damage, although low colchicine doses seem to have an excellent hepatic safety profile [70,71]. A meta-analysis published by Kedar et al. evaluating the efficacy and safety of colchicine showed no significant reduction in mortality risk [72]. Most data indicate no benefit from including colchicine in the standard treatment regimen in these patients.

Medical history, exclusion of other liver disorders using testing, and proof of injury associated with suspected therapeutic agents are required to diagnose DILI.

Several countries used hydroxychloroquine (HCQ) as one of the potential therapeutic strategies against COVID-19, despite the scarcity of scientific evidence and conflicting opinions regarding this drug [73,74]. Adverse musculoskeletal, hematological, cardiac, ophthalmological, and hepatic events are correlated with HCQ use [75]. However, severe liver dysfunction was rarely documented in the literature [76].

Corticosteroids have been widely administered during COVID-19 [77]. Prolonged use of these drugs can cause liver steatosis, while high doses can result in acute liver failure [78].

## 6. Clinical Correlations between SARS-CoV-2 Infection and Underlying Chronic Liver Disease

### 6.1. COVID-19 and Liver Cirrhosis

Ongoing studies are currently trying to determine the relationship between SARS-CoV-2 and chronic liver disease patients regarding their susceptibility to infection. To date, published studies have not suggested that patients with underlying chronic liver disease could have an increased susceptibility to SARS-CoV-2 infection [79], and United States medical data have indicated a reduced rate of positive testing among patients with liver cirrhosis [80,81]. However, chronic liver disease, including liver cirrhosis, is unlikely to protect against contracting SARS-CoV-2 infection. The lower rate of positive tests is probably due to strict adherence to prophylactic measures (e.g., social distancing and wearing a protective mask).

Patients with liver cirrhosis infected with SARS-CoV-2 show a gradual increase in morbidity and mortality related to the severity of liver disease, assessed by the Child–Pugh class. Thus, an increase in mortality was observed in patients with cirrhosis of Child–Pugh C, whose survivals decrease to 10% once subjected to mechanical ventilation. COVID-19 disease-related mortality was significantly associated with the severity of underlying liver cirrhosis, with the risk of death increasing in parallel with the severity class of liver disease: CP-A class 1.90%, CP-B 4.14%, and CP-C 9.32%, according to ongoing studies [82]. Strong evidence for increased disease severity in patients associated both COVID-19 and cirrhosis was presented in a large meta-analysis evaluating clinical data obtained from over 900,000 patients [83].

Cirrhosis disrupts both the local immunity of the liver and systemic immunity. This immune impairment may account for susceptibility to severe forms of COVID-19 and grave outcomes observed in this group [84].

Although acute mortality in patients with liver cirrhosis and COVID-19 is increased, in those patients who survive the initial episode, the risk of death or readmission at 90 days is similar to the risk observed in patients with liver cirrhosis without COVID-19 [85]. Therefore, except for the acute infectious period, the SARS-CoV-2 condition does not appear to precipitate liver disease progression.

On the other hand, other studies using multivariable analysis have published results depicting no correlation between cirrhosis and COVID-19 mortality [86,87]. A recent study performed by Simon et al. in 2021 did not find a specific relation between COVID-19 and the outcome or clinical course of cirrhosis [88].

### 6.2. COVID-19 and MAFLD

The etiology of liver diseases could influence the clinical evolution of SARS-CoV-2 infection. Risk factors associated with higher morbidity and mortality rates in SARS-CoV-2 infection are represented by age, obesity, and diabetes. However, there are inconsistencies in the literature regarding the influence of metabolic-associated fatty liver disease (MAFLD) on the clinical course of the SARS-CoV-2 condition, related to the difficulties in the differential diagnosis of MAFLD and other metabolic comorbidities [89,90].

Some reports describe a strong association between MAFLD and the progression of SARS-CoV-2 infection. Some studies show that people with MAFLD have an increased risk of symptomatic SARS-CoV-2 disease, a higher risk of progression to severe forms, and a longer time for viral clearance [91]. In MAFLD, the polarization status of macrophages could be disrupted, modulating inflammatory response to SARS-CoV-2 [91].

A meta-analysis of available data shows that among MAFLD patients with SARS-CoV-2 infection, obesity increases the risk of severe SARS-CoV-2 disease. These findings support the specific role of MAFLD in modulating susceptibility to SARS-CoV-2 infection and progression (Table 1) [92].

**Table 1.** Association between obesity and COVID-19 infection severity in patients with MAFLD (adapted from Asemota et al., 2022 [92]).

|  | OR | 95% CI | *p* |
|---|---|---|---|
| Unadjusted | 5.77 | 1.19–27.91 | 0.029 |
| Model I | 6.25 | 1.23–31.71 | 0.027 |
| Model II | 6.32 | 1.16–34.54 | 0.033 |

Model I: adjusted for age and gender. Model II: adjusted for age, gender, smoker status, type II diabetes mellitus, arterial hypertension, and dyslipidemia.

MAFLD is associated not only with COVID-19 but also with other systemic disorders, such as hyperglycemia, insulin resistance, altered immune status, obesity, vitamin D deficiency, diverticulosis, and anemia of chronic disease, through systemic inflammation [93–95].

### 6.3. COVID-19 and Chronic Hepatitis

A study of 1193 patients by Ronderos et al. [96] showed that chronic hepatitis C was correlated with increased in-hospital mortality in acute SARS-CoV-2 infection. Patients with chronic hepatitis C might have an increased risk of severe respiratory complications without previous comorbidity or COVID-19 liver damage [97]. The effects are correlated with the extrahepatic manifestations of HCV infection, which stimulate ACE2/TMPRSS mechanisms, and endothelial dysfunction and are secondary to the inflammatory process. However, more available data are needed for a clear conclusion.

A study conducted on 2482 patients by Kang et al. [98] documented the paradoxical relation between chronic hepatitis B and SARS-CoV-2 infection—the condition does not increase the risk of developing severe forms of COVID-19 and does not negatively influence disease outcomes, even if it appears that preexisting B virus infection and treatment with antiviral agents have a protective effect, decreasing the risk of contracting SARS-CoV-2

infection. The outcome of SARS-CoV-2 infection in HBV patients depends on the previous stage of chronic liver disease, described by Shanshan Yang et al. in a large study [99].

### 6.4. COVID-19 and Autoimmune Hepatitis

Many authors have focused their studies on the evolution of autoimmune hepatitis (AIH) in patients with SARS-CoV-2 infection. It was postulated that immunosuppressive treatment of AIH would increase these patients' risk of SARS-CoV-2 infection. Still, a prevalence of severe forms of COVID-19 was not observed, probably due to the prevention effect of a systemic inflammatory response by immunosuppressive therapy. Conversely, immunosuppressive treatment increases the time of SARS-CoV-2 virus clearance, and these patients become a source of contamination for a prolonged period [100].

An international multicentric study by Efe et al. related to outcomes of COVID-19 in patients with autoimmune hepatitis revealed that patients with AIH were not at higher risk for a worse prognosis with COVID-19 than other related causes of CLD [101].

A recent study, documented by Ashley L. Faulx et al. in 2021, suggested that AIH does not determine a more severe prognosis in co-infection with COVID-19, even in those patients receiving immunosuppressive drugs; thus, immunosuppressive treatment should not be interrupted in patients with AIH who develop severe forms of COVID-19, as there are no conclusive indications of worsening the clinical outcome in these patients [102].

Other studies on the relation between immunosuppressive medication and the prognosis of COVID-19 in patients diagnosed with AIH have concluded that systemic glucocorticoids or immunosuppressive therapy prescribed before the onset of COVID-19 was significantly associated with COVID-19-increased severity in patients with AIH [103].

Neeraj Kumar et al. describe a case of severe evolution of COVID-19 in a young man with autoimmune hepatitis, considering that morbidity in COVID-19 associated with liver disease is due to hyperinflammation and cytokine storm with increased IL-6 levels. Significant cytokine release and inflammatory responses affect both the onset and severity of disease progression. Consequently, a rapid and appropriate diagnostic evaluation and accurate estimation and prognosis are necessary [104].

### 6.5. COVID-19 and Vascular Diseases

A study conducted by Baiges et al. on patients with SARS-CoV-2 infection and underlying vascular liver diseases, including Budd–Chiari syndrome, portosinusoidal vascular disease and noncirrhotic splanchnic vein thrombosis, revealed a higher risk of SARS-CoV-2 infection and a higher risk of severe forms of COVID-19 [105]. Further studies in this area are needed.

## 7. COVID-19 and Cholangitis

Secondary sclerosing cholangitis is a chronic condition characterized by progressive fibrosis and biliary tract destruction, which can lead to biliary cirrhosis.

Post-COVID-19 cholangiopathy is a unique concept, defined as a variant of secondary sclerosing cholangitis. It can be determined by SARS-CoV-2 infection, or it can be drug induced. Cholangiopathy may be present in many other associated diseases, such as AIDS, cholangiolithiasis, diffuse intrahepatic metastases, and histiocytosis C [106].

The molecular mechanism can be explained due to the predominance of the ACE2 receptor in cholangiocytes. The presence of viral receptors on the host cell's surface significantly determines viral tropism. The penetration of SARS-CoV-2 in the host's cells is mediated by the S protein, which specifically interacts with ACE2 and transmembrane serine protease 2 (TMPRSS2) receptors. ACE2 expression is relatively low in hepatocytes and is significantly increased in cholangiocytes, while transmembrane serine protease 2 expression is higher in hepatocytes.

The binding of SARS-CoV-2 to the ACE2-receptors in cholangiocytes affects the barrier and the biliary acid transport mechanism by affecting gene regulation, leading to cholestasis [107].

Diagnosis depends on history, clinical evaluation, biological tests, and imaging studies.

In a study of 2047 patients admitted in hospital with COVID-19, 12 patients with severe COVID-18 developed cholangiopathy syndrome characterized by cholestasis and biliary tract abnormalities similar to the particularities observed in patients with secondary sclerosing cholangitis.

Histologic features included inflammation, strictures, cholangiocyte injury with microvascular anomalies, and fibrosis periportal hepatocytes metaplasia [106].

Biliary tract disorder in COVID-19 patients can be suspected when clinical and biological tests reveal cholestasis and elevated liver enzymes. Biliary imaging methods confirm the diagnosis of secondary sclerosing cholangitis.

Further research is needed to assess the pathogenesis in cholangiopathy associated with SARS-CoV-2 infection and to find preventive and optimal therapeutic measures [53].

## 8. COVID-19 and High-Risk Groups

According to their immunocompromised status, liver transplant recipients have an increased risk of severe clinical forms of SARS-CoV-2 infection. A retrospective study conducted by Colmenero et al. in hospitalized patients with liver transplants and COVID-19 highlighted a mortality rate of 18%, lower than in the general population, despite the evidence of a more severe disease outcome [108,109]. Similar results were cited in another multicentric study on 112 patients conducted by Rabbie et al., showing a mortality rate of 22.3%, lower than in patients without liver transplants [110].

This is due to these patients' immunomodulatory therapy, which can improve the systemic inflammatory response, reducing mortality [109,111]. However, immunosuppressive treatment can delay viral clearance, arguing for more severe clinical evolution [107]. An increased risk of severe forms and exitus in transplanted patients was correlated with young age, metabolic syndrome association, prescription of antibiotics and vasopressor treatment [110].

Pregnant women are a high-risk population category for severe outcomes of SARS-CoV-2 infection, with an increased prevalence of premature births. Among important maternal mortality causes are HELLP syndrome, hemolysis, liver cytolysis and preeclampsia. The studies carried out support an increased risk of preeclampsia in pregnant women with SARS-CoV-2 infection, suggesting a possible overlap of physiopathological hypotheses [112].

Risk factors correlated with severe forms of COVID-19 in pregnant women include smoking, obesity, diabetes, and preeclampsia [113,114].

Although pregnancy is not associated with increased susceptibility to contracting SARS-CoV-2 infection, pregnant women are at higher risk of severe outcomes [112,115] and complications, such as preeclampsia or HELLP syndrome [112,116,117]. Thrombocytopenia, increased liver enzymes and hemolysis are markers present both in HELLP syndrome and in patients with multiorgan dysfunction in critical condition [112], suggesting that SARS-CoV-2 infection has similarities with HELLP syndrome in pregnant women [109] and raises real challenges in the differential diagnosis. It is assumed that there is a pathogenetic link between SARS-CoV-2 and HELLP syndrome, an association that requires extensive future research.

## 9. Conclusions

Liver damage in SARS-CoV-2 infection plays an essential role in COVID-19 disease. This is clinically significant, especially in patients with pre-existing liver diseases that have a higher risk of severe COVID-19 and death.

Systemic inflammation, coagulation disorder, endothelial damage, and immune dysfunction explain the pathogenic mechanisms involved in the deterioration of liver function.

Although various mechanisms of action of the SARS-CoV-2 virus on liver cells have been studied, the clinical impact of the direct viral effect on liver cells has not been estab-

lished. Some severe COVID-19 cases have reported acute liver damage associated with increased mortality.

Immune dysfunction related to liver cirrhosis has a much more harmful effect on the clinical outcome of SARS-CoV-2 infection than medication-induced immunosuppression in these patients.

More extensive observational studies are required to establish the impact of COVID-19 infection on liver function.

**Author Contributions:** Conceptualization, C.M.M., C.M.V., E.C. and P.M.; methodology, M.P.; software, A.O.D. and R.M.; validation, C.M.M., P.M. and C.M.V.; formal analysis, C.M.M.; investigation, V.B.; resources, C.M.M. and M.P.; data curation, M.S.P.; writing—original draft preparation, C.M.M. and E.C.; writing—review and editing, C.M.V.; visualization, C.M.M.; supervision, P.M. All authors have read and agreed to the published version of the manuscript.

**Funding:** This research received no external funding.

**Conflicts of Interest:** The authors declare no conflict of interest.

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
