# Peer review of "Features of Liver Injury in COVID-19 Pathophysiological, Biological and Clinical Particularities"

_gastroent, doi:10.3390/gastroent14020012_

Round 1

Reviewer 1 Report

Overall the authors cover almost every aspect of liver involvement in COVID-19. However, I think that a lot of references are lacking and so, the consumptions made by the authors are misleading.

1. In pathophysiology and histology section, it is true that ACE-2 is mainly expressed in hepatocytes, however in liver damage (and thus liver regeneration) is expressed both in cholangiocytes as well as hepatocytes (ref: Paizis G, Tikellis C, Cooper ME, Schembri JM, Lew RA, Smith AI, et al. Chronic liver injury in rats and humans upregulates the novel enzyme angiotensin converting enzyme 2. Gut 2005;54:1790-1796) in order for the liver to achieve faster regeneration. That is very important for patiens with cirrhosis where the internalization of ACE-2 due to SARS-CoV-2 reduces its availability on the cell surface and could lead to aggravated fibrosis and deterioration of liver function, leading to acute on chronic liver failure.

2. Figure 2 isn't supported by robust data and should be deleted since it could mislead clinicians

3.  In line 194, the authors write that anticoagulants are a "documented cause of drug-induced liver damage". They base this assumption in a paper by Mahamid M from 2011 with 2 patients with RA experiencing DILI. Firstly this paper is irrelevant. Secondly, low molecular heparins, broadly used in patients with severe COVID-19 are a rare cause of DILI, as can be seen in LiverTox, which should be cited instead (LiverTox: Clinical and Research Information on Drug-Induced Liver Injury [Internet]. Bethesda (MD): National Institute of Diabetes and Digestive and Kidney Diseases; 2012-. Low Molecular Weight Heparins. [Updated 2017 Nov 13]. Available from: https://www.ncbi.nlm.nih.gov/books/NBK548009/ and Hahn KJ, Morales SJ, Lewis JH. Enoxaparin-Induced Liver Injury: Case Report and Review of the Literature and FDA Adverse Event Reporting System (FAERS). Drug Saf Case Rep. 2015 Dec;2(1):17)

4. In line 202, colchicine is discussed as COVID-19 treatment. Though with a favorable adverse events profile, multiple metanalyses have shown no benefit, as has the PRINCIPLE and RECOVERY trials. It is also worth noting that even COLCORONA trial showed moderate benefit of colchicine use. As a result colchicine is no longer recommended for SARS-CoV-2 infection.

5. In the next section, the first lines discuss the possibility of SARS-CoV-2 infection in patients with cirrhosis. This part is irrelevant and should be ommited. Instead it should be emphasized the fact that cirrhosis, especially when decompensated, comes with a significant mortality and COVID-19 related complications. Moreover, it should be discussed that the lack of antibody responses in these patients might act as an aggravating factor for severe COVID-19. In this section more references are needed. 

6. In HCV section , it should be noted that although a few studies show that active HCV replication is correlated with worse outcomes, there is paucity of data in these patients

7. The cholangiitis part should be moved somwhere else and certainly not in the "pre-existing liver diseases" section, unless the authors want to discuss pre-existin PBC or PSC, where data are scarce

8. In the AIH section, the papers by Efe and Marjot should be included and the contradictory results of the 2 studies by Efe discussed (ref:  Marjot T, Buescher G, Sebode M, Barnes E, Barritt AS 4th, Armstrong MJ, et al. SARS-CoV-2 infection in patients with autoimmune hepatitis. J Hepatol 2021;74:1335-1343,   Efe C, Dhanasekaran R, Lammert C, Ebik B, Higuera-de la Tijera F, Aloman C, et al. Outcome of COVID-19 in Patients With Autoimmune Hepatitis: An International Multicenter Study. Hepatology 2021;73:2099-2109, Efe C, Lammert C, TaÅŸçılar K, Dhanasekaran R, Ebik B, Higuera-de la Tijera F, et al. Effects of immunosuppressive drugs on COVID-19 severity in patients with autoimmune hepatitis. Liver Int 2022;42:607-614). 

9. English language should be carefully re-checked and various mistakes corected

Author Response

Dear reviewer,

We highly appreciate you consideration and suggestions. We tried to address all your requierements.

Q1 .In pathophysiology and histology section, it is true that ACE-2 is mainly expressed in hepatocytes, however in liver damage (and thus liver regeneration) is expressed both in cholangiocytes as well as hepatocytes (ref: Paizis G, Tikellis C, Cooper ME, Schembri JM, Lew RA, Smith AI, et al. Chronic liver injury in rats and humans upregulates the novel enzyme angiotensin converting enzyme 2. Gut 2005;54:1790-1796) in order for the liver to achieve faster regeneration. That is very important for patiens with cirrhosis where the internalization of ACE-2 due to SARS-CoV-2 reduces its availability on the cell surface and could lead to aggravated fibrosis and deterioration of liver function, leading to acute on chronic liver failure.

A1 . We highlighted in our paper the evidence of ACE2 expression and its up-regulation in human cirrhotic liver,  described by Paizis G et al, and its role in the deterioration of liver function, reference [9].

Q2. Figure 2 isn't supported by robust data and should be deleted since it could mislead clinicians

A2. At your suggestion, we removed Figure 2 from the paper.

Q3. In line 194, the authors write that anticoagulants are a "documented cause of drug-induced liver damage". They base this assumption in a paper by Mahamid M from 2011 with 2 patients with RA experiencing DILI. Firstly this paper is irrelevant. Secondly, low molecular heparins, broadly used in patients with severe COVID-19 are a rare cause of DILI, as can be seen in LiverTox, which should be cited instead (LiverTox: Clinical and Research Information on Drug-Induced Liver Injury [Internet]. Bethesda (MD): National Institute of Diabetes and Digestive and Kidney Diseases; 2012-. Low Molecular Weight Heparins. [Updated 2017 Nov 13]. Available from: https://www.ncbi.nlm.nih.gov/books/NBK548009/ and Hahn KJ, Morales SJ, Lewis JH. Enoxaparin-Induced Liver Injury: Case Report and Review of the Literature and FDA Adverse Event Reporting System (FAERS). Drug Saf Case Rep. 2015 Dec;2(1):17)

A3. We revised the considerations of drug-induced liver injury caused by low molecular weight heparins, broadly used in patients with severe COVID-19, mentioned the references provided, and deleted the outside source.

Q4. In line 202, colchicine is discussed as COVID-19 treatment. Though with a favorable adverse events profile, multiple metanalyses have shown no benefit, as has the PRINCIPLE and RECOVERY trials. It is also worth noting that even COLCORONA trial showed moderate benefit of colchicine use. As a result colchicine is no longer recommended for SARS-CoV-2 infection.

A4. We updated the information on colchicine use, to reflect current data regarding the lack of efficacy on COVID-19 mortality, adding new references.

Q5.In the next section, the first lines discuss the possibility of SARS-CoV-2 infection in patients with cirrhosis. This part is irrelevant and should be ommited. Instead it should be emphasized the fact that cirrhosis, especially when decompensated, comes with a significant mortality and COVID-19 related complications. Moreover, it should be discussed that the lack of antibody responses in these patients might act as an aggravating factor for severe COVID-19. In this section more references are needed.

A5. We updated the section referring to the impact of COVID-19 on patients with liver cirrhosis, adding the results of other studies that emphasized the risk of severe disease related to underlying liver cirrhosis, and improved our references , at yout suggestion.

Q6: In HCV section , it should be noted that although a few studies show that active HCV replication is correlated with worse outcomes, there is paucity of data in these patients.

A6: We specified that available data is insufficient for a clear conclusion regarding the impact of COVID-19 on HCV patients.

Q7: The cholangitis part should be moved somwhere else and certainly not in the "pre-existing liver diseases" section, unless the authors want to discuss pre-existin PBC or PSC, where data are scarce

A7: Thank you for pointing out this mistake. We created a new section (chapter 7) for discussing the relationship between SARS-CoV-2 infection and cholangiopathy.

Q8: In the AIH section, the papers by Efe and Marjot should be included and the contradictory results of the 2 studies by Efe discussed (ref:  Marjot T, Buescher G, Sebode M, Barnes E, Barritt AS 4th, Armstrong MJ, et al. SARS-CoV-2 infection in patients with autoimmune hepatitis. J Hepatol 2021;74:1335-1343,   Efe C, Dhanasekaran R, Lammert C, Ebik B, Higuera-de la Tijera F, Aloman C, et al. Outcome of COVID-19 in Patients With Autoimmune Hepatitis: An International Multicenter Study. Hepatology 2021;73:2099-2109, Efe C, Lammert C, TaÅŸçılar K, Dhanasekaran R, Ebik B, Higuera-de la Tijera F, et al. Effects of immunosuppressive drugs on COVID-19 severity in patients with autoimmune hepatitis. Liver Int 2022;42:607-614).

A8: We included in our paper the contradictory results of the studies conducted by Efe et al, discussing the relation between AIH, immunosuppressive treatment and the outcomes of COVID-19 in these patients. We added the mentioned references.

Q9: English language should be carefully re-checked and various mistakes corected

A9: We proofread and a native speaker performed an English revision.

Reviewer 2 Report

1. Was this review invited or not? If not invited than this review should be either scoping review or systematic review. In any case methodology section is missing and it must be added in order to answer the following questions: a) how did you choose the literature you cited? b) what was the database searched; c) what key words did you use?

2. Entire paper is very long. For example- lines 171-179 is very descriptive without providing much information. The entire paper should be cut by at least 20% , and especially introduction section. References should remain extensive as this is a review paper. 

3. References are not properly placed in the text. You should put references , for example, in line 184- can cause liver damage [64]. not " can cause liver damage. [64]". This needs to be changed throughout the text

4. hydroxychloroquine and steroids are another causes of DILI and needs to be mentioned

5. Line 269- diverticulosis is associated with NAFLD and should be added here: 10.3390/medicina58010038

6. Did authors find any studies examining COVID risk and complication in patients with vascular liver diseases? 

7. Did authors find any study examining risk in patients with hepatocellular carcinoma and liver metastasis from other cancers?

8. NAFLD nomenclature has changed recently to MAFLD and authors might consider changing it in their review 10.1053/j.gastro.2019.11.312

Author Response

Dear reviewer,

Thank you so much for your constructive suggestions and your consideration!

Q1: Was this review invited or not? If not invited than this review should be either scoping review or systematic review. In any case methodology section is missing and it must be added in order to answer the following questions: a) how did you choose the literature you cited? b) what was the database searched; c) what key words did you use?

A1: Yes, there was an invitation to write a review. Our paper is a literature review nor a systematic or meta-analysis review. According to this, adding a Materials and Methods section is irrelevant.

Q3: References are not properly placed in the text. You should put references , for example, in line 184- can cause liver damage [64]. not " can cause liver damage. [64]". This needs to be changed throughout the text

A3: Thank you for pointing out this mistake. We changed and replaced the references properly throughout the text.

Q4: hydroxychloroquine and steroids are another causes of DILI and needs to be mentioned

A4: We updated the DILI chapter (Chapter 5) with information regarding hydroxychloroquine and corticosteroid use, adding the corresponding references.

Q5: Line 269- diverticulosis is associated with NAFLD and should be added here: 10.3390/medicina58010038

A5: We added diverticulosis to the list and provided the appropriate citation. Thank you for your suggestion.

Q6: Did authors find any studies examining COVID risk and complication in patients with vascular liver diseases?

A6: We mentioned an increased risk of severe outcomes of COVID-19 in patients with vascular liver disease (portosinusoidal vascular disease, noncirrhotic splanchnic vein thrombosis, and Budd Chiari syndrome), providing adequate references.

Q7: Did authors find any study examining risk in patients with hepatocellular carcinoma and liver metastasis from other cancers?

A7:We consider that our manuscript is not focusing in assessing the risk in patients with hepatocellular carcinoma and liver metastasis from other cancer. Such a broad topic should be treated separately, and further studies regarding the particular condition of these patients are needed.

Q8: NAFLD nomenclature has changed recently to MAFLD and authors might consider changing it in their review 10.1053/j.gastro.2019.11.312

A8: We revised chapter 5.2 using the new nomenclature of NAFLD.

Round 2

Reviewer 1 Report

I think that after the review the manuscript is ready for publication

Author Response

Thank you.

Reviewer 2 Report

The authors responded that methodology section is irrelevant. I respectfully disagree. In my opinion this is a crucial part of the paper. The authors are not expert on this topic so narrative review without methodology section, and literature selection is mandatory. 

Author Response

Dear reviewer,   We added a Materials and Methods chapter (Chapter 2) at your suggestion, documenting the methodology used to select included articles. This review aims to make a comprehensive and integrated approach to the essential aspects of liver involvement in COVID-19, including pathogenic mechanisms, biochemical abnormalities, and correlations between preexisting liver diseases and the effects of SARS-CoV- 2. A series of electronic searches in PubMed was conducted using the following keywords: "COVID-19", "SARS-CoV-2", “chronic liver disease”, “cirrhosis”, “drug-induced liver injury”,  and "NAFLD”. Articles obtained from these search queries were manually assessed for information quality and results' significance. References cited in selected articles were also assessed and included if deemed relevant.

Round 3

Reviewer 2 Report

I appreciate the authors attention in adding methodology section. The paper has been adequately improved and I do not have further comments